# Exploring RGB+Depth Fusion for Real-Time Object Detection

**DOI:** 10.3390/s19040866

**Published:** 2019-02-19

**Authors:** Tanguy Ophoff, Kristof Van Beeck, Toon Goedemé

**Affiliations:** EAVISE, KU Leuven, 2860 Sint-Katelijne-Waver, Belgium; kristof.vanbeeck@kuleuven.be (K.V.B.); toon.goedeme@kuleuven.be (T.G.)

**Keywords:** Object detection, Single-shot, Neural Networks, Sensor fusion, RGB, Depth, RGBD

## Abstract

In this paper, we investigate whether fusing depth information on top of normal RGB data for camera-based object detection can help to increase the performance of current state-of-the-art single-shot detection networks. Indeed, depth sensing is easily acquired using depth cameras such as a Kinect or stereo setups. We investigate the optimal manner to perform this sensor fusion with a special focus on lightweight single-pass convolutional neural network (CNN) architectures, enabling real-time processing on limited hardware. For this, we implement a network architecture allowing us to parameterize at which network layer both information sources are fused together. We performed exhaustive experiments to determine the optimal fusion point in the network, from which we can conclude that fusing towards the mid to late layers provides the best results. Our best fusion models significantly outperform the baseline RGB network in both accuracy and localization of the detections.

## 1. Introduction

Fast and highly accurate object detection is a key ingredient for a manifold of applications. An example is the detection of objects in various industrial applications such as pick and place machines or safety applications such as pedestrian collision avoidance systems for autonomous vehicles. For the latter application, the demands are very stringent, both with respect to the detection accuracy as well as in its real-time behavior. The current state-of-the-art literature describes highly accurate detection systems. However, there are still several issues that remain to be solved. For example, detecting individual objects in a group, or more generally detecting highly occluded objects, are some of the remaining problems current detection systems are facing. Besides providing highly accurate detections, many applications require these detectors to process images in real-time. Important steps have been taken towards real-time detection systems with single-pass convolutional neural networks (CNN). However, a weakness these real-time detection networks have, is the difficulty to provide accurate bounding boxes around the detected objects. Optimal localization is crucial for further segmentation or re-id applications.

Looking back, traditional computer vision techniques used hand-crafted features as input to a learned classifier to perform various tasks such as object detection and classification [1,2,3]. In order to increase the detection accuracy, these techniques have been successfully extended using features from various different sources like thermal imaging [4,5] and depth [6,7,8]. Benenson et al. [9] also proved that depth can be a strong cue for preprocessing images, reducing the detection search space.

However, with the rise of CNNs, deep learning has been outperforming traditional computer vision techniques by a significant margin [10,11]. Object detection is often based on a combination of a CNN for classification with a sliding-window or other region proposal approach [12,13]. While these techniques have proven to be successful, a major drawback is that they are highly computationally expensive. These techniques generate a great number of region proposals, of which each needs to be evaluated by a classification network. These issues have been partially solved by sharing computations between the region proposal and classification pipelines [14,15].

Finally, the current state-of-the-art for fast object detection lies with single-pass detection networks, such as YOLO [16,17,18] and SSD [19]. By re-framing object detection as a single regression problem, one can use a single network which simultaneously outputs bounding box coordinates and class scores. Beside the fact that these networks are orders of magnitude faster than previous work, they also have the advantage of considering the entire image when computing bounding boxes (i.e., exploiting context information). This additional information makes for even better detectors, outperforming previous detection pipelines.

While these state-of-the-art detection networks have achieved impressive results for object detection, there are still several issues that remain to be solved. Taking inspiration from traditional computer vision techniques, some of these issues have been solved by combining these detectors with sensor fusion [20,21,22,23]. For example, adding extra information in the form of thermal imaging allowed these detectors to perform better in challenging day-and-night scenarios [23].

The combination of depth information with normal RGB data has already been successfully used with CNNs to improve the accuracy on the classification task [24,25]. These techniques work by running two networks for each data stream and only fuse the features at the end of the networks, effectively performing late fusion. Initial steps have been taken towards the use of RGBD data with CNN-based object detection [26,27], proving the added value of depth for this task as well. However, they still consist of a suboptimal multi-stage approach, and only fuse the different features at the end of their pipeline.

In previous work [28], we investigated the added value of RGB+Depth fusion for real-time pedestrian detection systems, by fusing RGB and depth data in a single-shot end-to-end network. We proved that depth is an interesting channel for pedestrian detection, as it provides a more simple depiction of the environment with basic silhouettes of the persons. This allows a network to detect individual objects more easily and helps to provide more accurate bounding boxes. However, this previous work was naive in the exact position of the fusion in the network. Only few specific points were considered in the network architecture to perform this sensor fusion. No emphasis was given to the trade-off between fusion point and optimal accuracy. Moreover, it was only experimentally verified that the arbitrary midway fusion was best for a specific object detection task, *i.c.* pedestrian detection.

Therefore, in this paper we take an extensive look at where in the network to best perform this sensor fusion. To the best of our knowledge, no exhaustive search has been reported in literature of where the optimal level is to fuse the two information streams, both for the case of RGB+Depth as well as for the case of RGB+IR. Apart from this, we also validate our solution on object detection in general and on different datasets with different methods of depth acquisition.

To summarize, our main contributions are:We propose a flexible fusion layer architecture that can be placed at any arbitrary level of a neural network. We implemented a single-stage detection network for RGB+Depth sensor fusion, with a parameterizable fusion level. This model was made available in our open-source PyTorch library, Lightnet (https://www.gitlab.com/eavise/lightnet). We performed brute-force experiments to determine the optimal fusion location in our network, by exhaustively training a model for every fusion level in our network. We validated the potential of sensor fusion to improve the performance of object detectors on multiple datasets for diverse objects in different situations, both concerning detection accuracy and localization.

## 2. Materials and Methods

We assume the input of our algorithm consists of a sensor that both acquires an RGB image, as well as a depth image (D). Both are assumed to be aligned with each other, yielding a 4-channel RGBD image. The latter can easily be realized with an RGBD range camera based on e.g., structured light, time-of-flight or a calibrated stereo setup. In the remainder of this section we will discuss our network architecture, the training methodology that we employed and our evaluation metrics.

### 2.1. Network Architecture

We opted to base our architecture on the YOLOv2 architecture, a state-of-the-art single-pass detection network [17]. This network achieves a very impressive speed-accuracy trade-off and as such is able to perform detections on embedded devices like the NVIDIA Jetson TX2 in real-time [17,28]. It consists of 27 layers, which gives us 28 different possibilities to fuse our information streams, being right before the first layer and after every of the 27 layers of the network.

As seen in Figure 1a, our fusion network first has two subnetworks, of which one processes our main RGB channels and one the D channel. Both subnetworks have exactly the same architecture, consisting of the first *N* layers of YOLOv2, but have different weights to process their respective inputs. After these *N* separate layers, we fuse both streams together by means of concatenation, and then continue our network for the last 27−N layers of the YOLOv2 detector.

In our previous work, where we only naively fused at specific places in the network, we adapted the convolutions where the fusion happened in the network [28]. As this becomes infeasible to do for every different permutation of our fusion network, we created a special fuse layer. This layer easily enables the parameterization of the fusion level (see Figure 1b). The fuse layer first concatenates the feature maps of our two subnetworks, after which it performs a 1×1 convolution on this combined feature map. The practical purpose of this convolution is to reduce the number of output feature maps of both concatenated streams back to the original number of feature maps in one network. This allows the fuse layer to be placed anywhere in the network in a transparent manner. We also believe this convolution to be beneficial for fusing both streams together. By combining together the feature maps, the convolution is able to extract features that were represented in both the main and fusion subnetworks and create a stronger combined feature map set than both sets individually.

### 2.2. Training Methodology

We trained and evaluated these models on three highly diverse application cases (see Section 3), following the same methodology for both of them.

First we trained a regular RGB detection network, as well as a depth-only network. These models give us a baseline to compare our results, as well as a way to tune our hyperparameters. Most of the hyperparameters are identical to the default YOLOv2 parameters, but the length of the training was changed to take into account the size of the dataset. As is customary when training object detection networks, we start from pretrained weights and perform transfer learning on our datasets. YOLOv2, which is based on the darknet-19 classification network, uses pretrained weights from this classification network on the ImageNet dataset [29], up to its 23th layer. For the RGB network we followed the exact same procedure. Since depth only consists of one channel and thus has a different first convolutional layer, it is not optimal to use these pretrained weights. However, as there are no equivalent ImageNet datasets available for depth, we decided to use the RGB pretrained weights, but removed the weights of the first layer. The motivation for this is that the network looks for similar features in both the depth maps and RGB images, and thus will still benefit from RGB pretrained weights over randomly initialized weights.

We then proceeded with the training of the 28 different fusion networks. For this purpose, we used exactly the same training hyperparameters as for the RGB network. This ensured that we did not fine-tune the hyperparameters for the fusion networks better as compared to the RGB baseline, resulting in a fair comparison of the different models. However, this implies that our fusion models, which have more parameters to train, might perform suboptimally.

We employed the same procedure of transfer learning and used these same ImageNet pretrained weights on our RGB subnetwork, as well as on the main network after the fusion, up to its 23th layer (see Figure 2). For the depth subnetwork we faced the same problem as explained above, but decided not to use the same pretrained RGB ImageNet weights without first layer. The fusion networks have more parameters and thus might have a more difficult time to converge. On top of that, if the depth subnetwork does not provide any substantial information compared to the RGB network, the fusion layer could possibly ignore those feature maps. Therefore we decided to use the weights from our previously trained depth-only network as pretrained weights for the depth subnetwork. These weights will already be trained to extract meaningful information from the depth maps and would thus be a better starting point to training our fusion network.

As an additional experiment we kept the weights after the fusion layer randomly initialized. Our motivation for this was to reduce the bias of the network towards the RGB features. However, our experiments indicated that the performance significantly decreased for the earlier fusion levels, since this means that those models have less layers with pretrained weights.

### 2.3. Evaluation Metrics

To compare our different networks we use the traditional precision-recall curves *(PR)* and average precision scores *(AP)*. These metrics give us a global overview of the accuracy of a network, as well as insight in where the network fails (False Positives and False Negatives). We count an object as properly detected if the detection bounding box overlaps the annotation bounding box with an intersection over union (IoU) threshold of at least 50%.

One of our main hypotheses about the advantages of fusing depth maps is the fact that the clearly distinguishable silhouettes in the depth maps allow for more accurate bounding boxes around the detected objects. To evaluate this, we measure the AP of our networks, using the relatively new COCO IoU thresholding scheme [30], which is defined as follows.
(1)$$AP_{COCO}=\frac{\sum_{IoU\in I}AP_{IoU}\left(Annotations,Detections\right)}{I};I=\left\{0.50,0.55,0.60,\ldots ,0.95\right\}$$

This means the metric rewards techniques that offer a better localization and is thus a good fit for proving our hypothesis.

## 3. Results

We performed exhaustive experiments on three different application cases: pedestrian detection, multi-class road user detection and the detection of screws on PCBs. The first application case is evaluated using a dataset which consists of Kinect RGBD images of persons. For the second application case we use stereo RGB images of cars, cyclists and pedestrians. Hence, this enables us to investigate the influence of the quality of the depth acquisition on the fusion potential. The third case has more of an industrial context. It consists of grayscale and depth images from printed circuit boards (PCB) in which the goal is to detect screws. This case is included to proof that our RGB and depth fusion easily generalizes itself to highly diverse scenarios.

### 3.1. Kinect Pedestrian Detection

We first evaluated our new networks on the application of pedestrian detection (see Figure 3). For this, we used the EPFL Pedestrian Dataset [31], which we recently relabeled [28]. This dataset consists of around 5000 RGBD images of persons, captured with a Kinect V2 and calibrated with the color camera. We split this dataset in the same train, validation and test set as explained in [28]. We trained our networks for 40,000 batches of 64 images and compared the different networks on our test set (see Figure 4).

Figure 4a displays the AP metric versus the position of the fusion layer for our different networks. First and foremost, we note that almost all of our fusion networks outperform our RGB baseline, with midway to late fusion giving the best results with a maximal increase of 3.3%. This demonstrates that object detection indeed benefits from using depth information on top of regular RGB data.

It can also be observed that our depth-only network outperforms the RGB network as well. This might point to the fact that depth is even more meaningful than RGB for this dataset. Indeed, when looking at the example images in Figure 3b, we can clearly distinguish persons from the silhouettes in the depth maps only. Despite its good performance, we note that 14 of the fusion networks offer an even better detection accuracy than the pure depth network, proving the advantage of sensor fusion in this case. The best accuracy is reached at the 14th fusion level, indeed supporting our previous naive claim [28] that midway fusion is optimal. Figure 4a also shows a best fit parabola to remove training-induced noise, indicating that a theoretical optimal fusion level in this case is located around level 18.

The AP_COCO_, depicted in Figure 4b, follows the same trend. The difference between our RGB baseline and best fusion network is 3.8%, which indicates that the silhouettes in the depth maps help the networks to provide better located bounding boxes. This can also be seen in the example images in Figure 3, where we see that the fusion detections provide better bounding boxes around the persons, without cutting of parts of the head or feet.

### 3.2. Multi-Class Road User Detection Based on Stereo Depth

To further investigate the added value of depth on top of normal RGB for multi-scale road users detection, we also evaluated our networks on the KITTI dataset [32]. This dataset consists of around 7500 stereo RGB images, captured from a car driving around a city and has annotations for cars, cyclists and pedestrians. To compute depth maps from this stereo data, we used the stereoSGBM [33] algorithm from OpenCV [34] (see Figure 5b). Since the test set of this dataset is part of a benchmark, the annotations are not publicly available. We thus split the original training set in a training and validation set, and will report the results of our networks on this validation split. We train a single network for the three different classes and train each network with the default YOLOv2 configuration, because the lack of a different validation and test set does not allow us to fine-tune the hyperparameters.

We followed the same procedures for evaluation as explained in the KITTI benchmark paper [32]. This means an IoU overlap of 70% for cars, and 50% for cyclists and pedestrians is required to account for a true positive detection. We computed an AP metric for each of the three difficulty levels described in the benchmark (easy, moderate, hard) and also computed a mean AP (mAP), by averaging the results of the different classes (see Table 1). Note that, as opposed to the previous application case, we do not perform AP_COCO_ metric evaluations for this dataset, since the different IoU settings for the various classes results in an unfair comparison.

Figure 6 shows the different AP values for the moderate difficulty, which is the main evaluation metric of the benchmark. These bar charts show the same tendency as for the previous dataset. Overall, our fusion models outperform the baseline RGB detector and the parabolas show the optimal fusion point to be around level 15. We do note that the results for the car class are quite a bit higher than the results of the other classes. This is most likely due to the imbalance of class instances in this dataset, as there are 28,742 cars, 1627 cyclists and 4487 pedestrians in the images and we did not account for that (e.g., by weighing our loss function for the different classes). However, we do not think this imbalance influences the interpretability of our results to prove the added value of depth fusion on top of normal RGB. Indeed, this experiment proves that this fusion approach is also beneficial if the quality of the depth is not ideal, which is mostly the case for depth computed from stereo images (for instance, depth cannot be calculated on image regions with limited texture). Moreover, in this experiment we only used a very basic stereo depth estimation technique based on block matching, which is nowadays greatly surpassed by CNN-based techniques.

### 3.3. Localization of Screws on PCBs

In order to validate the generalizability of our methodology on more diverse objects and situations, we evaluated our fusion networks on an industrial dataset consisting of grayscale and depth images of electronic devices, with the aim of detecting the screws on them [35]. For this, we use the GD Screws dataset provided by the Life Cycle Engineering lab of KU Leuven (see Figure 7). The images in this dataset are acquired using a sheet-of-light scanner, which provides highly accurate depth maps, better than the Kinect in the previous case. However, this dataset provides only grayscale images instead of RGB.

The size of this dataset is limited (75 images), and as such we decided to perform a random train and test split of 75% and 25%. Without validation set there is again no justifiable manner to tune the hyperparameters. We therefore used the default YOLOv2 values, and only altered the length of the training to 2000 batches of 16 images each.

Looking at Figure 8, we observe the same trend as with our previous experiments. Globally, our fusion networks outperform our Grayscale baseline, with our best fusion network having an increase in AP of 1.0%. The smaller gain compared to the previous datasets is most likely due to the fact that the scores are higher overall and thus the room for improvement is smaller.

When looking at the AP_COCO_, we conclude that depth indeed helps our fusion networks provide better bounding boxes, as the maximal increase here is 4.9%. Screws being small objects means that providing accurate bounding boxes is more difficult. The bigger increase in the AP_COCO_ again validates our hypothesis that the depth maps allow the network to provide better bounding boxes.

## 4. Discussion

Our experiments indicate that RGB+Depth fusion offers significant benefits to the detection accuracy, which we measured using the AP metric. The different datasets all have different ways of acquiring the depth maps, demonstrating that our architecture can be used in a multitude of applications and setups. The fitted parabolas show that a mid to late fusion level performs better than earlier levels of fusion. This is likely due to the fact that lower level features (such as edges and basic shapes) might not always align perfectly between our main and fusion channels. In our first and second dataset this is mainly caused due to different lens distortions and camera calibrations. In the last dataset we detect different features in both information source: e.g., the specific disc-shape of the screw in the grayscale image and the black border around the screws in the depth maps. However, the deeper we advance through the network, the more abstract the features become and the more we spatially sub-sample through max-pooling layers. This allows the network to more easily cope with the misalignment in matching features between both information sources, leading to better results for later fusion levels.

The AP_COCO_ metrics demonstrate our claim that depth maps offer a way to provide better localized bounding boxes. The increase in AP_COCO_ score between our RGB baseline and the fusion models is indeed more pronounced than the regular AP metric for most cases. This implies that our fusion models perform better compared to the baseline when increasing the IoU threshold on the evaluation metric, thus validating our hypothesis. This can also be seen in the example images, where the bounding boxes have a better fit around the detected objects.

## 5. Conclusions

In this paper we proposed the fusion of RGB+Depth images using single-pass detection networks. We performed exhaustive experiments, in order to determine the best level to fuse both information sources. For this purpose, we constructed a fuse layer architecture, capable of fusing both streams at any arbitrary layer in the network. This layer—as well as the complete detection architecture—has been added to our open-source PyTorch library, Lightnet.

Our experiments show that RGB+Depth fusion increases both the general detection accuracy and the localization performance of the bounding boxes, regardless of the depth acquisition method. Furthermore, our results seem to indicate that mid to late fusion performs best, though there is no exact pronounced optimal fusion level. As such, the fusion level could be an additional hyperparameter, which should be tuned separately for every different case.

In the future, we aim to investigate whether more complex fusion mechanisms could further improve the results of our models. Indeed, using different or more convolutions in the fuse layer, or adding non-linearities in the form of activation functions might improve the performance of our fusion networks even more. Another possibility would be to look at whether different representations of the depth maps, like three-channel colorized depth would improve the results as well. In the case of stereo, we could also fuse both camera viewpoints, effectively skipping the computation of the depth map.

## Figures and Tables

**Figure 1 sensors-19-00866-f001:**
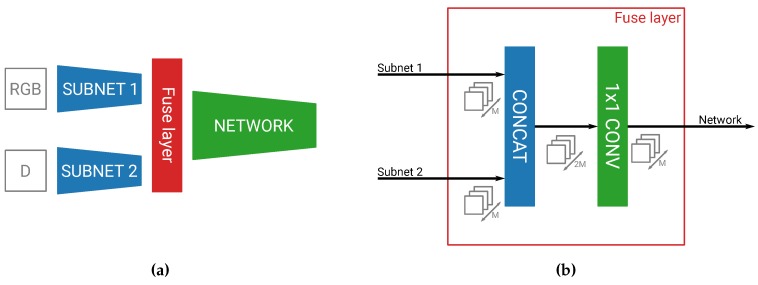
The main building blocks of our parameterizable fusion network. (**a**) The fuse layer can be transparently implemented after any arbitrary layer, allowing for a parameterizable fusion level. (**b**) The fuse layer combines both information streams and divides the number of output channels by two.

**Figure 2 sensors-19-00866-f002:**
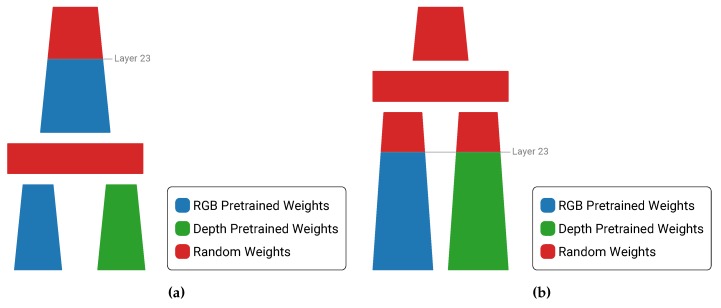
The pretrained weights we used when starting the training of our fusion networks. (**a**) Pretrained weights when fusing before convolutional layer 23. (**b**) Pretrained weights when fusing after convolutional layer 23.

**Figure 3 sensors-19-00866-f003:**
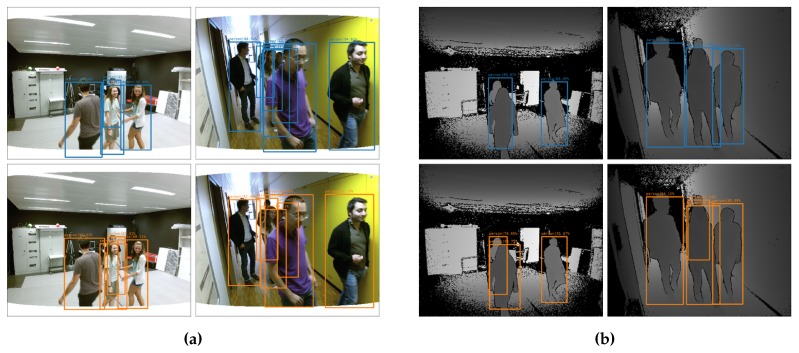
Example images from the test set of the Relabeled EPFL Pedestrian dataset. Detections from our best fusion model (RGBD_18) are shown in comparison with our baseline RGB and D models. (**a**) Resulting detections of our RGB model (top images—blue boxes) and best fusion model (bottom images—orange boxes). (**b**) Resulting detections of our Depth model (top images—blue boxes) and best fusion model (bottom images—orange boxes).

**Figure 4 sensors-19-00866-f004:**
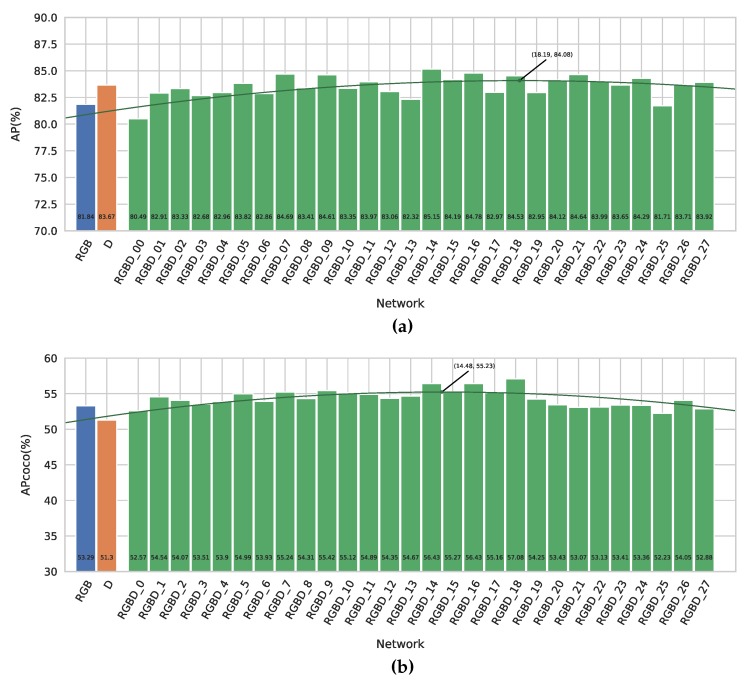
The results of our different networks on the Relabeled EPFL Pedestrian Dataset. (**a**) The regular AP metric of our different networks. (**b**) The AP_COCO_ metric of our different networks.

**Figure 5 sensors-19-00866-f005:**
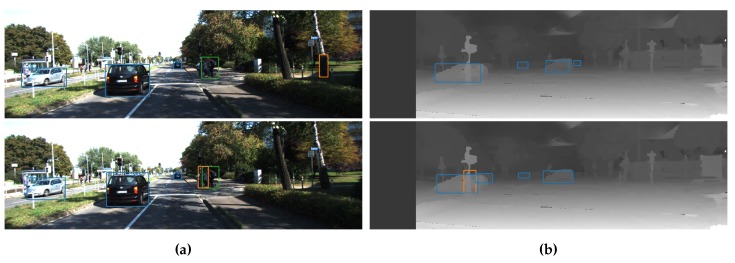
Example images from our validation set of the KITTI dataset. Detections from our best fusion model (RGBD_08) are shown in comparison with our baseline RGB and D models. (Cars—blue; Cyclists—green; Pedestrians—orange). (**a**) Resulting detections of our RGB model (top image) and best fusion model (bottom image). (**b**) Resulting detections of our Depth model (top image) and best fusion model (bottom image). Note that we increased the brightness and contrast of these images for clarity.

**Figure 6 sensors-19-00866-f006:**
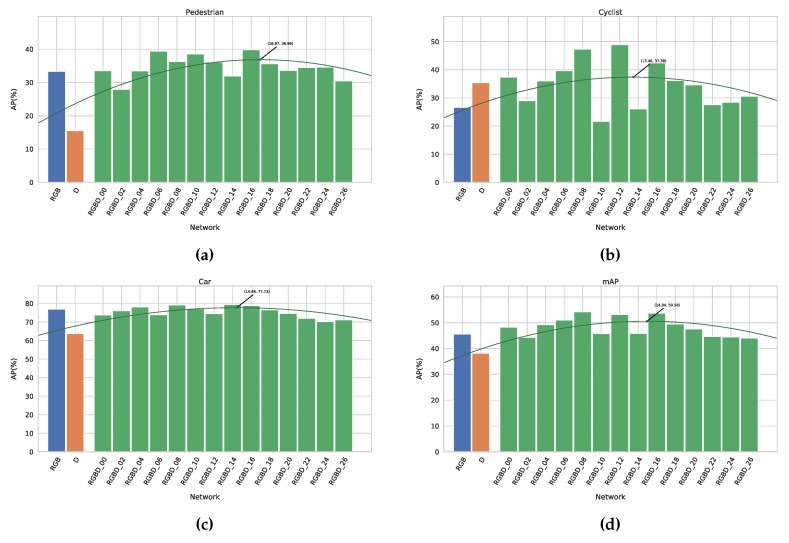
The regular AP metric of our different networks on the KITTI dataset (moderate difficulty). (**a**) Pedestrian class. (**b**) Cyclist class. (**c**) Car class. (**d**) mAP.

**Figure 7 sensors-19-00866-f007:**
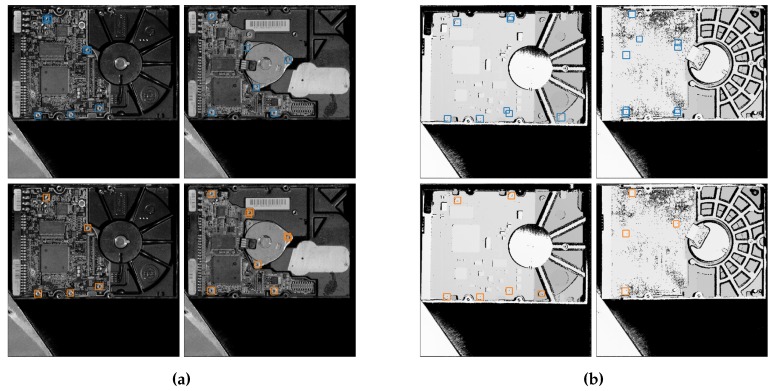
Example images from the test set of the GD Screws dataset. Detections from our best fusion model (GD_21) are shown in comparison with our baseline Grayscale and D models. (**a**) Resulting detections of our Grayscale model (top images—blue boxes) and best fusion model (bottom images—orange boxes). (**b**) Resulting detections of our Depth model (top images—blue boxes) and best fusion model (bottom images—orange boxes).

**Figure 8 sensors-19-00866-f008:**
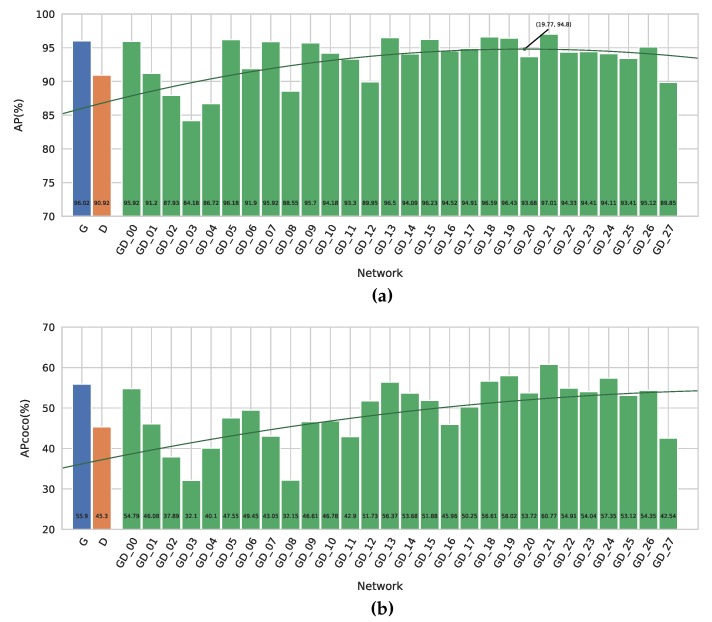
The results of our different networks on the GD Screws Dataset. (**a**) The regular AP metric of our different networks. (**b**) The AP_COCO_ metric of our different networks.

**Table 1 sensors-19-00866-t001:** AP values of our different models on the KITTI dataset. The best results are highlighted in red.

Class Difficulty	Pedestrian			Cyclist			Car			mAP		
	Easy	Moderate	Hard	Easy	Moderate	Hard	Easy	Moderate	Hard	Easy	Moderate	Hard
RGB	41.66	33.31	25.62	40.01	26.62	28.26	88.12	76.98	65.73	56.6	45.64	39.87
D	12.12	15.47	12.24	57.06	35.36	33.24	83.88	63.68	54.1	51.02	38.17	33.19
RGBD_00	40.92	33.52	25.69	52.83	37.39	37.08	82.77	73.84	63.64	58.84	48.25	42.14
RGBD_02	29.73	27.91	21.92	45.46	28.99	28.46	86.6	76.13	65.45	53.93	44.34	38.61
RGBD_04	33.98	33.44	25.75	52.22	36.04	34.37	84.2	78.06	67.08	56.8	49.18	42.4
RGBD_06	41.41	39.44	30.54	55.43	39.66	38.57	84.9	73.86	63.66	60.58	50.99	44.26
RGBD_08	43.29	36.3	28.16	71.82	47.31	47.78	87.86	79.15	68.53	67.66	54.25	48.16
RGBD_10	40.78	38.54	29.67	31.11	21.63	20.0	90.24	77.13	66.25	54.05	45.77	38.64
RGBD_12	39.42	36.12	28.07	70.46	48.92	48.26	84.2	74.55	64.04	64.69	53.2	46.79
RGBD_14	38.07	31.94	25.08	38.8	26.05	24.88	87.03	79.46	68.38	54.63	45.81	39.45
RGBD_16	49.86	39.83	31.29	63.31	42.44	41.38	90.79	78.84	68.5	67.98	53.7	47.06
RGBD_18	41.11	35.6	27.8	56.51	36.18	33.63	85.91	76.54	65.98	61.18	49.44	42.47
RGBD_20	39.89	33.59	25.73	54.08	34.58	33.35	86.53	74.65	64.24	60.17	47.61	41.11
RGBD_22	36.55	34.44	26.62	43.74	27.55	25.37	86.13	71.99	61.85	55.47	44.66	37.95
RGBD_24	38.65	34.59	26.81	48.9	28.47	26.93	85.09	70.26	60.48	57.55	44.44	38.07
RGBD_26	33.13	30.47	24.56	47.22	30.58	28.67	86.13	71.13	61.27	55.49	44.06	38.17

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
