# Peer review of "Exploring RGB+Depth Fusion for Real-Time Object Detection"

_sensors, 2019, doi:10.3390/s19040866_

Round 1

Reviewer 1 Report

This paper explores the use of RGBD data for object detection using deep learning. The problem is interesting and contributions wellcome. 

The idea of using deep learning on RGBD data is not new, though. I think that the introduction should reflect this state of the art in the introduction

Another general comment about the paper is that the algorithm is applied for person detection and screw detection. While this sounds interesting at first, one gets a bit dissapointed with the dataset used for screw detection there are too few examples to conclude anything. At least authors should use any kind of cross-validation. In any case, I think that the screws part does not contribute significantly to the focus of the paper and the application of person detection could be enough.

I also think that the paper could also be improved with more details about the  model structure when using fusion (fig 1), for instance are subnets 1,2 exactly the same layers as in yolo up to layer N? then the layers in "network" green layer are exactly the same layers of the original yolo?

Regarding the first layer of the depth channel, the authors say that in this case they remove the RGB weights. Another simple option is to convert the gray scale image to color-gray-scale by repeating the depth channel 3 times. 

The paragrapth between lines 124-133 is confusing, I needed to read it several times to understand how the network is initialzed. I think that this could be improved with a table, figure....

Author Response

Dear reviewer,

We would like to thank you for the time you spent reviewing our paper and for the constructive feedback you provided. We read your review and refactored the paper to accommodate for your insightful comments and questions. We highlighted all textual changes in yellow in our revised manuscript for your convenience.

Hereby our point-by-point response to your review:

The idea of using deep learning on RGBD data is not new, though. I think that the introduction should reflect this state of the art in the introduction

Indeed, RGBD data has already been used for eg. classification and detection networks.

We added an extra paragraph in the introduction (L49-55) to reflect this.

Another general comment about the paper is that the algorithm is applied for person detection and screw detection. While this sounds interesting at first, one gets a bit dissapointed with the dataset used for screw detection there are too few examples to conclude anything. At least authors should use any kind of cross-validation. In any case, I think that the screws part does not contribute significantly to the focus of the paper and the application of person detection could be enough.

Sadly we did not have enough time to perform some kind of cross-validation, although we do admit the screws dataset to be somewhat small. However, seeing as this is a completely different setting, we decided to leave this part in the paper as we do believe it shows our method can be used for different applications with different methods of acquisition.

To further demonstrate the usability of our architectural design, we decided to validate our networks on the widely used and publicly available KITTI dataset (Section 3.2). Our experiments show that depth also improves upon the RGB baseline for stereo depth and works for different classes (cars, cyclists, pedestrians).

As you probably know, training networks takes quite some time and as such we were not able to train all 30 networks in these 10 days. We thus decided to only try fusion on the even layers to reduce the number of networks to train to 16. We think the 2 other datasets already show the consistency in our results and fusing every other layer in this new dataset is enough to show a trend for these results.

I also think that the paper could also be improved with more details about the  model structure when using fusion (fig 1), for instance are subnets 1,2 exactly the same layers as in yolo up to layer N? then the layers in "network" green layer are exactly the same layers of the original yolo?

Yes, these are the same layers as YOLOv2.

Seeing as this was not entirely clear, we slightly changed our text to be more explicit (L95-97).

Regarding the first layer of the depth channel, the authors say that in this case they remove the RGB weights. Another simple option is to convert the gray scale image to color-gray-scale by repeating the depth channel 3 times. 

This is indeed a valid point. Another option to investigate would be to look at colorized depth instead of the single-channel depth we have been using. As we do not have the time to investigate all these options right now, we added this as future work (L272-274).

The paragrapth between lines 124-133 is confusing, I needed to read it several times to understand how the network is initialzed. I think that this could be improved with a table, figure...

Upon re-reading the paragraph ourselves, we completely agree! We slightly rewrote this paragraph (L130-137) and added a figure (Fig. 2) in the hopes of making this clearer for future readers.

Reviewer 2 Report

Manuscript ID sensors-421712

Title Exploring RGB+Depth Fusion for Real-Time Object Detection

Authors Tanguy Ophoff * , Kristof Van Beeck , Toon Goedemé

Review Date: 23 January 2019

This paper presents experiments on fusion of depth data into layers of convolutional neural networks (CNNs) used for object detection.  The main contribution is the analysis of when fusion is most beneficial, by brute-force experimentation on fusing the depth data at every possible layer.  The results conclude that mid-to late fusion has the highest benefit.

Overall the paper itself is very well written.  The writing, grammar, and flow are very nice and the paper is easy to understand and follow.

However, in its current state, this reviewer is hesitant to endorse this paper for several reasons.

The literature review, while nicely covering a broad history of object detection approaches, is shallow or entirely missing when it comes to direct state of the art in RGB-D CNN object detection.  

The experiments consist of two datasets - one from the authors' prior work and another from the authors' institution.  While RGB-D datasets are not as common as others typically used in CNNs, there do exist datasets on which the authors could compare their results.  This is the biggest problem with the paper - there is no real comparison against other competing methods except against the authors' prior work.  Overall, this work seems quite preliminary - this may still be of value as long as it's more clearly articulated.

Lastly, there could be further analysis and discussion (along with references to relevant literature) to make an argument why the results provided are convincing.

Minor comments:

-Please be careful using strong words like "extensive experiments" and "thus proving our hypothesis".  These words may overestimate the significance of the work presented in the paper. 

- Check/edit the firstname.lastname email address in the author list.

Examples of some relevant related work include, but not limited to:

A. Eitel, J. T. Springenberg, L. Spinello, M. Riedmiller and W. Burgard, "Multimodal deep learning for robust RGB-D object recognition," 2015 IEEE/RSJ International Conference on Intelligent Robots and Systems (IROS), Hamburg, 2015, pp. 681-687.

W. Choi, C. Pantofaru and S. Savarese, "Detecting and tracking people using an RGB-D camera via multiple detector fusion," 2011 IEEE International Conference on Computer Vision Workshops (ICCV Workshops), Barcelona, 2011, pp. 1076-1083.

M. Schwarz, H. Schulz and S. Behnke, "RGB-D object recognition and pose estimation based on pre-trained convolutional neural network features," 2015 IEEE International Conference on Robotics and Automation (ICRA), Seattle, WA, 2015, pp. 1329-1335

Gupta S., Girshick R., Arbeláez P., Malik J. (2014) Learning Rich Features from RGB-D Images for Object Detection and Segmentation. In: Fleet D., Pajdla T., Schiele B., Tuytelaars T. (eds) Computer Vision – ECCV 2014

Available dataset example:

E.g. K. Lai, L. Bo, X. Ren, D. Fox, "A large-scale hierarchical multiview rgb-d object dataset", Proc. of the IEEE Int. Conf. on Robotics & Automation (ICRA), 2011.)

Author Response

Dear reviewer,

We would like to thank you for the time you spent reviewing our paper and for the constructive feedback you provided. We read your review and refactored the paper to accommodate for your insightful comments and questions. We highlighted all textual changes in yellow in our revised manuscript for your convenience.

Hereby our point-by-point response to your review:

The literature review, while nicely covering a broad history of object detection approaches, is shallow or entirely missing when it comes to direct state of the art in RGB-D CNN object detection.  

You are completely correct and we thank you a lot for the relevant related works.

We added an extra paragraph in the introduction (L49-55) to reflect this comment and referenced the 4 papers you provided (L28, L50, L53).

The experiments consist of two datasets - one from the authors' prior work and another from the authors' institution.  While RGB-D datasets are not as common as others typically used in CNNs, there do exist datasets on which the authors could compare their results.  This is the biggest problem with the paper - there is no real comparison against other competing methods except against the authors' prior work.  Overall, this work seems quite preliminary - this may still be of value as long as it's more clearly articulated.

The main point of our paper is to prove the added value of Depth fusion on top of normal RGB for object detection, and not to beat the state-of-the-art detection on a certain dataset. We decided to start from YOLOv2 for internal reasons (ic. wanting a fast detector for embedded systems), but it should not matter what base detector you choose as the idea of fusing depth should be implementable in almost any detection network.

The first dataset is also not from our own work, it is a public dataset published by the EPFL. We reworked the annotations as they were lacking in some images, and added extra information about the occlusion and truncation of the persons. These annotations are also publicly available for anyone to experiment with. Your confusion can be understood, as we accidentally forgot to add a citation to the original EPFL paper (It was in our bibliography, but we somehow forgot to add the actual citation in the text). This has now been fixed on line 167.

However, we do agree that we needed to evaluate on some bigger dataset. As the example dataset you kindly provided consisted of some staged situations, we decided to instead use the KITTI dataset, which has real street scenarios and is in our opinion a more useful dataset for our technique. Our experiments show that depth also improves upon the RGB baseline for stereo depth and works for different classes (cars, cyclists, pedestrians).

As you probably know, training networks takes quite some time and as such we were not able to train all 30 networks in these 10 days. We thus decided to only try fusion on the even layers to reduce the number of networks to train to 16. We think the 2 other datasets already show the consistency in our results and fusing every other layer in this new dataset is enough to show a trend for these results.

Sadly we could not compare directly with the leaderboards of the KITTI dataset, as they only allow one submission per person (to prevent fine-tuning on the test set). However, we observe that our resulting technique reaches a speed vs. accuracy trade-off comparable to other state-of-the-art methods on the leaderboard (with the results on our validation set).

You can find the details of our new experiments in section 3.2.

Lastly, there could be further analysis and discussion (along with references to relevant literature) to make an argument why the results provided are convincing.

As further analysis, we included the extra KITTI dataset, as explained above. The addition of an extra dataset, with a more realistic scenario and yet another depth acquisition method further demonstrates the potential of depth fusion on top of normal RGB.

Please be careful using strong words like "extensive experiments" and "thus proving our hypothesis".  These words may overestimate the significance of the work presented in the paper. 

We changed 'extensive' to 'exhaustive' and 'brute-force' (L76, L157) and changed 'proving' to 'validating' (L256), as we indeed understand your concern. We hope this change in wording clearly depicts our original intent whilst writing the paper.

Check/edit the firstname.lastname email address in the author list

Thank you for your comment. We changed it accordingly.

Round 2

Reviewer 2 Report

The authors have sufficiently addressed the comments from the original review in their revised submission.  Overall this is a nicely presented paper.  The additional experiments on the KITTI dataset strengthen the paper and help address some of the main concerns in the experiments. 

The paper could still be stronger with a larger literature review of the most relevant methods, deeper experimentation, and further analysis of the results that compares against other depth fusion work.